Methods

# A high-throughput two-cell assay for interrogating inhibitory signaling pathways in T cells

Sumana Sharma[1],*, Toby Whitehead[1],*, Mateusz Kotowski[1], Emily Zhi Qing Ng[1], Joseph Clarke[1], Judith Leitner[2], Yi-Ling Chen[1], Ana Mafalda Santos[1], Peter Steinberger[2], Simon J Davis[1]

**The recent success of immunotherapies relying on manipulation of T-cell activation highlights the value of characterising the mediators of immune checkpoint signaling. CRISPR/Cas9 is a popular approach for interrogating signaling pathways; however, the lack of appropriate assays for studying inhibitory signaling in T cells is limiting the use of large-scale perturbation-based approaches. Here, we adapted an existing Jurkat cell-based transcriptional reporter assay to study both activatory and inhibitory (PD-1-mediated) T-cell signaling using CRISPR-based genome screening in arrayed and pooled formats. We targeted 64 SH2 domain-containing proteins expressed by Jurkat T cells in an arrayed screen, in which individual targets could be assessed independently, showing that arrays can be used to study mediators of both activatory and inhibitory signaling. Pooled screens succeeded in simultaneously identifying many of the known mediators of proximal activating and inhibitory T-cell signaling, including SHP2 and PD-1, confirming the utility of the method. Altogether, the data suggested that SHP2 is the major PD-1-specific, SH2 family mediator of inhibitory signaling. These approaches should allow the systematic analysis of signaling pathways in T cells.**

## Introduction

T cells have a key role in immune-mediated host protection from cancer, infection, and autoimmune diseases. Activation of T cells requires ligand binding by cell surface receptors, which initiate downstream signal transduction cascades (Smith-Garvin et al, 2009). These signaling events mostly rely on tyrosine phosphorylation of the cytosolic tails of the surface receptors, which in turn allows recruitment of intracellular signaling proteins that contain phosphotyrosine (pTyr)-binding domains. Many of the proteins involved in this process contain one or more Src homology 2 (SH2)

domains, 100 amino-acid structures that bind specifically to pTyr. Signaling is initiated by the T-cell receptor (TCR), which binds peptides presented by APCs in the form of complexes with MHC molecules, at contacts depleted of large phosphatases where receptor phosphorylation by the SH2 domain-containing tyrosine kinase LCK is proposed to be favoured (Davis & van der Merwe, 2006). Either CD4 or CD8 co-receptors can also bind the MHC, stabilizing the association of LCK with the phosphorylated TCR (Jiang et al, 2011). TCR/CD3 complex phosphorylation allows the recruitment of a key downstream effector, ZAP-70, another SH2 domain-containing protein, to CD3ζ. Phosphorylation of the membrane-anchored LAT protein by ZAP-70 at multiple Tyr sites results in turn in protein recruitment and formation of a signaling complex comprising of multiple SH2 domain-containing proteins including GADS, GRB2, SOS, VAV1, ITK, and PLC-γ1, leading eventually to the initiation of transcription by NFκB1, NFAT, and AP-1 (Courtney et al, 2018; Shah et al, 2021).

Although the signaling pathway proximal to the TCR has been very well characterised, one aspect of T-cell signaling that is yet to be fully explored is how inhibitory signals are generated, although the involvement of yet more SH2 domain-containing proteins is clear (Lorenz, 2009). Two SH2 domain-containing phosphatases, SHP1 and SHP2, also known as PTPN6 and PTPN11, respectively, are thought to be involved in inhibitory signaling downstream of many inhibitory receptors including two of the most important, PD-1 and BTLA. These phosphatases limit T-cell activation by dephosphorylating key TCR signaling molecules including ZAP-70 and CD3ζ (Sheppard et al, 2004). SHP1 was initially believed to be the main effector downstream of both receptors (Riley, 2009), although more recent work suggests that PD-1 primarily recruits SHP2, and BTLA SHP1, as a result of sequence differences in their pTyr motifs (Celis-Gutierrez et al, 2019; Xu et al, 2020, 2021). On the other hand, inhibitory effects of both proteins have also been proposed to be only partially because of the activities of SHP1 and SHP2, implying that inhibitory signaling by these receptors is yet to be fully characterised (Xu et al, 2020).

Recently, multiple studies have employed unbiased genome-wide screening approaches, using CRISPR/Cas9-based gene editing

[1]MRC Translational Immune Discovery Unit, John Radcliffe Hospital, University of Oxford, Oxford, UK   [2]Division of Immune Receptors and T Cell Activation, Institute of Immunology, Medical University of Vienna, Vienna, Austria

Correspondence: sumana.sharma@rdm.ox.ac.uk; simon.davis@imm.ox.ac.uk
*Sumana Sharma and Toby Whitehead contributed equally to this work

in both primary and immortalised T-cell lines, to unravel cellular mechanisms regulating T-cell proliferation, cytotoxicity, tissue infiltration, cell differentiation, and cell signaling (Shang et al, 2018; Shifrut et al, 2018; Kotowski & Sharma, 2020; Belk et al, 2022). Large-scale activation screens have previously involved T cells being activated with anti-CD3 antibody that is either plate- or bead-bound, or in solution, rather than stimulation being applied in the setting of more physiological cell–cell contacts. The advantage of a cell-based activation assay, especially, is that it allows free diffusion of receptors at cell–cell contacts, allowing signal integration. Receptor/ligand pairings can easily be introduced exogenously into the system, so that the integration of co-stimulatory and co-inhibitory signals can be studied systematically. An assay of this type has previously been established by Jutz et al., who used it to characterize the functions of both activatory and inhibitory receptors (Jutz et al, 2017) using fluorescence-based transcriptional reporters expressed in the human Jurkat T cell line. In these assays, a T-cell stimulator (TCS) cell, which is a murine thymoma-derived cell line (BW5417) engineered to express a membrane-bound single-chain Fv (scFv) variant of the anti-CD3$\varepsilon$ antibody OKT3, is used as the APC (Leitner et al, 2010). TCS cells are versatile tools as they can be readily engineered to express a given surface protein to study T-cell co-inhibitory or co-stimulatory processes, hence this system provides an ideal platform to characterise signalling processes in T cells.

Here, we first adapt the cellular assay developed by Jutz et al (2017), by generating a Cas9 expressing version of the Jurkat cell reporter line to make it amenable to systematic arrayed and large-scale CRISPR-based "knock-out (KO)" screens. We illustrate how such an assay can be used to perform arrayed KO screens to individually characterise the roles of the 64 SH2 domain-containing proteins that are expressed by Jurkat T cells in activatory and inhibitory signaling. We then use the method to perform large-scale, genome-wide activation and inhibition screens in a bid to discover new regulators of T-cell signaling during activation via the TCR and inhibition by PD-1 upon engagement with PD-L1. In this way, we establish a method for the efficient perturbation-based interrogation of T-cell signaling in a physiological setting.

# Results

### A two-cell system for genetic analysis of inhibitory and activatory signaling in T cells

To study activatory T-cell signaling in the context of both co-activatory and co-inhibitory signals from CD28 and PD-1, we first established a range of signaling settings using Jurkat T cells in the two-cell assay system described by Leitner et al (2010). In this system, the Jurkat reporter cell lines, with or without PD-1, henceforth referred to as Jurkat/PD-1 and Jurkat/control, are stimulated with a previously described TCS, a murine thymoma cell line, BW5417, that co-expresses a membrane-bound single-chain Fv (scFv) variant of the anti-CD3$\varepsilon$ antibody OKT3 and high levels of PD-L1 and CD86 (TCS/CD86/PD-L1) (Leitner et al, 2010). The Jurkat cells are engineered to express eGFP under the control of the NF$\kappa$B

transcription factor, allowing eGFP levels to be used as a measure of activation (see schematics in Fig 1A). We sought to study inhibitory signaling, in a single cell-line, in the context of low activatory signaling, in which only the TCR/CD3 complex is engaged (signal 1), and high activatory signaling, wherein both TCR and CD28 are engaged (signal 1 + 2). To achieve this, we co-cultured Jurkat cells with stimulator cell line TCS/CD86/PD-L1 that were pretreated with (i) anti-CD86, (ii) anti-PD-L1 or (iii) both anti-CD86 and anti-PD-L1 blocking antibodies, creating stimulator cells equivalent to TCS/PD-L1, TCS/CD86, and TCS only, respectively. Cultures of Jurkat/control cells to which no TCS were added were used as baseline controls. In the signal 1 condition, Jurkat/control cells stimulated with TCS produced eGFP expression levels equivalent to those stimulated with TCS/PD-L1 cells, whereas Jurkat/PD-1 cells stimulated with TCS had significantly higher levels of eGFP expression compared with those stimulated with TCS/PD-L1 cells, demonstrating the inhibitory effects of PD-1 on the activatory pathway upon PD-L1 engagement (Fig 1B). In the signal 1 + 2 context (TCS/CD86), all the cells produced more eGFP, but PD-1 engagement (TCS/CD86/PD-L1) resulted in significantly less signaling (Fig 1B).

To enable genetic manipulation of the Jurkat cells used in the two-cell assay, we then generated a high-efficiency Cas9 version of both the Jurkat/control and Jurkat/PD-1 cell lines, using lentiviral transduction (Fig S1). We then transduced cells with single gRNAs targeting *NFKB1* and *RELA* and observed a decrease in the expression of eGFP when these cell lines were stimulated with TCS compared with control cells targeted with empty gRNA. This confirmed that eGFP production relied on the NF$\kappa$B signaling pathway and the two-cell system was amenable to manipulation with CRISPR-based perturbations using single gRNAs (Fig 1C). Next, given the known roles of SHP1 and SHP2 phosphatases in regulating inhibitory signaling, we targeted *PTPN6* (encoding SHP1) and *PTPN11* (encoding SHP2) using two separate gRNAs and stimulated the PD-1 expressing reporter lines in Signal 1 conditions (TCS and TCS/PD-L1). Targeting *PTPN11* resulted in an increase in signaling when PD-1 was engaged, whereas targeting *PTPN6* had no effect, reconfirming the role of SHP2 in mediating the inhibitory effects of PD-1 (Fig 1D). We then tested the two *PTPN11*-targeting guides in activation under signal 1 + 2 conditions (TCS/CD86 and TCS/CD86/PD-L1) and observed that targeting cells with the better-performing guide (*PTPN11*-g2) rescued signaling (Fig 1E). Altogether, these observations are consistent with recent work showing that PD-1 recruits SHP2 but not SHP1, and indicate that PD-1 inhibition is effective even in the absence of co-stimulatory signaling by CD28, presumably by acting directly on the TCR(/CD3) signaling pathway (Xu et al, 2021).

### Arrayed CRISPR-based screening of the role of SH2 domain-containing proteins in T-cell signaling

Having established the cell-based assay that allows signaling through the NF$\kappa$B pathway and the effects of PD-1 on this pathway to be studied, we used the assay to study the role of SH2 domain-containing proteins in regulating protein tyrosine kinase-mediated signaling in T cells. We compiled a list of 64 SH2 domain-containing proteins that are expressed in Jurkat cells, which included SHP1 and SHP2, based on transcriptome analysis of Jurkat cells from Cell Model Passport (van der Meer et al, 2019, Table S1). We created a

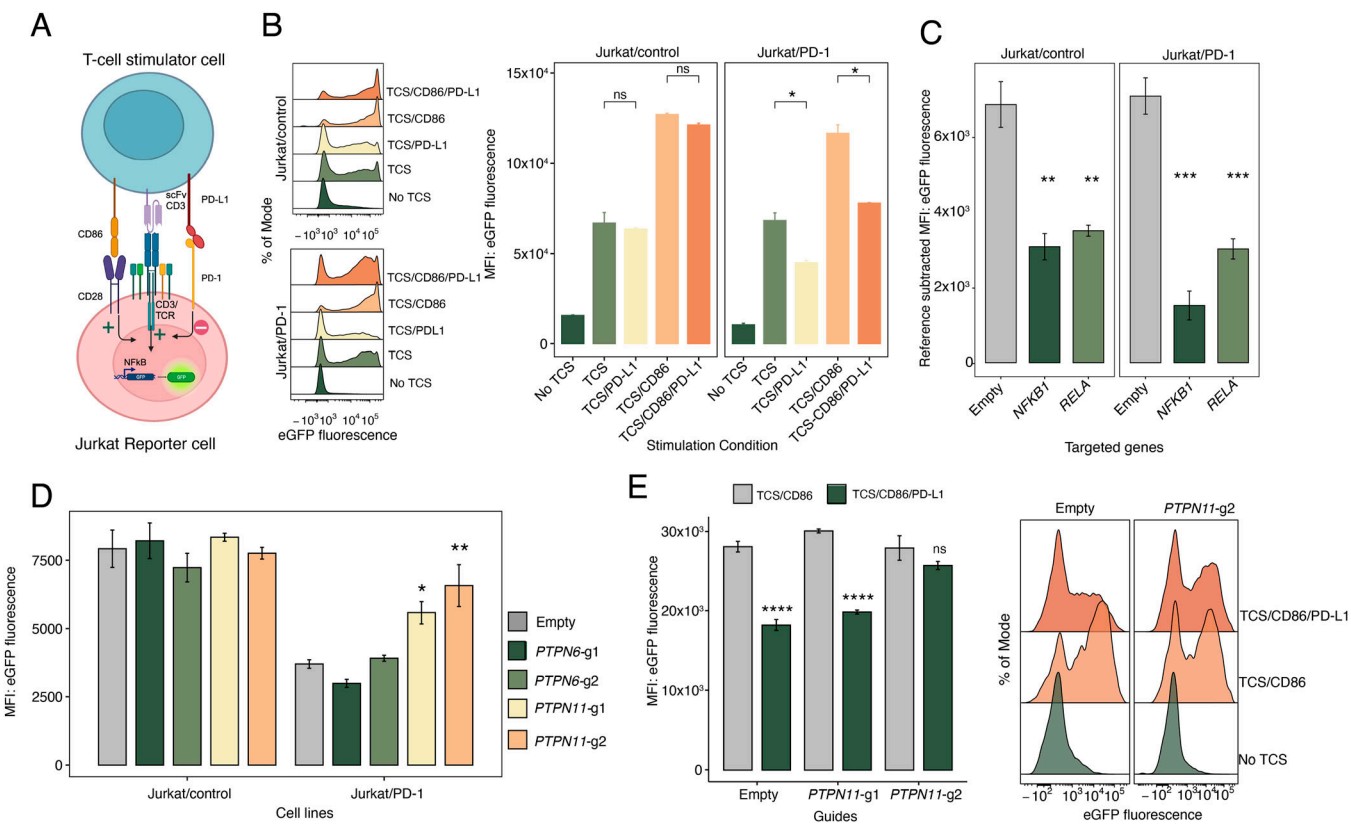

**Figure 1. An activation and inhibition assay amenable to single gRNA CRISPR-based genome editing.**
**(A)** Schematics of the cellular assay to study activatory and inhibitory signaling pathways in Jurkat T cells (created with BioRender.com). **(B)** Representative flow plots of eGFP fluorescence levels measured by flow cytometry of Jurkat/control and Jurkat/PD-1 cells co-cultured with four different T-cell stimulator (TCS) conditions or cultured without stimulation (left panel). Mean eGFP fluorescence intensity from three replicate experiments in Jurkat/control and Jurkat/PD-1 cells co-cultured from four different TCS conditions or cultured without stimulation (right panel). Pairwise comparisons performed using independent *t* test. **(C)** Mean eGFP fluorescence intensity from three replicate experiments in Jurkat/control and Jurkat/PD-1 cells targeted by empty, *RELA* or *NFKB1* sgRNAs after co-culture with TCS only. Pairwise comparisons performed using independent *t* test against empty-targeted samples within each Jurkat group. **(D)** Median eGFP fluorescence intensity from three replicate experiments in Jurkat/control and Jurkat/PD-1 cells targeted by empty, *PTPN6* or *PTPN11* sgRNAs after co-culture with TCS/PD-L1. Pairwise comparisons performed using one-way ANOVA, comparing against empty-targeted samples within each Jurkat group. **(E)** Median eGFP fluorescence intensity of Jurkat/PD-1 cells targeted by empty or *PTPN11* sgRNAs co-cultured with TCS/CD86 or TCS/CD86/PD-L1, and eGFP fluorescence levels of Jurkat/PD-1 cells targeted with the same sgRNAs in the same conditions or without stimulation. Pairwise comparison performed using independent *t* test between TCS/CD86 and TCS/CD86/PD-L1 stimulated samples (left panel). Representative plots of eGFP fluorescence levels measured by flow cytometry for empty and *PTPN11*-targeted Jurkat/PD-1 cell line unstimulated or stimulated with TCS/CD86 and TCS/CD86/PD-L1 (right panel). In all cases, the number of replicates is three and *P*-value asterisks represent: ****$P < 0.001$, ***$P < 0.001$, **$P < 0.01$, *$P < 0.05$; ns, not significant.

CRISPR-based KO arrayed library to target the remaining 62 SH2 domain-containing proteins with two gRNAs for each gene. We then used the two-cell assay to assess signaling outcomes in all five signaling contexts for all the "SH2-targeted" T cells (summarised in Fig 2A). For each stimulation condition, we then identified the targeted genes producing the most extreme changes in eGFP expression (Fig S2). To investigate how targeting of these genes affected signaling under different thresholds of activation and PD-1-mediated inhibitory conditions, we individually investigated each gene. We observed that *CSK*-, *CBL*-, and *SLA*-targeted cells exhibited increased background signaling in the absence of TCR/CD3 complex engagement and produced increased fold changes in eGFP fluorescence compared with the empty transduction control in all stimulation contexts (Fig 2B, upper panel). This confirmed the known role of CSK in phosphorylating, specifically, the LCK inhibitory tyrosine (Tyr505) to reduce the activity of the kinase. Loss of CSK expression allows LCK to remain active and phosphorylate the ITAMs of the TCR-associated CD3

chains, leading to activation without receptor ligation (Schoenborn et al, 2011). The ubiquitin ligase CBL is a well-known negative regulator of TCR signaling and its loss leads to reduced clearance of engaged TCR from the cell surface, thereby increasing TCR signals. SLA (also known as SLIP-1) is another known negative regulator of signaling by the TCR thought to act by linking ZAP-70 with CBL (Tang et al, 1999). Although stimulation of *CBL*- and *CSK*-targeted T-cells with TCS/PD-L1 cells under signal 1-type (TCS and TCS/PD-L1) conditions significantly increased activation levels compared with the non-targeted ("empty") control, both mutants produced lower eGFP levels when PD-1 was engaged. In contrast, under signal 1 + 2 conditions (TCS/CD86 and TCS/CD86/PD-L1), the inhibitory effects of PD-1 were lost for all three targeted genes, indicating that activating signals can readily overwhelm even potent inhibitory signaling mediators (Fig 2B).

Next, we investigated the genes whose targeting led to reduced activation in all signaling contexts (Fig 2A), that is, *NFKB1* (positive

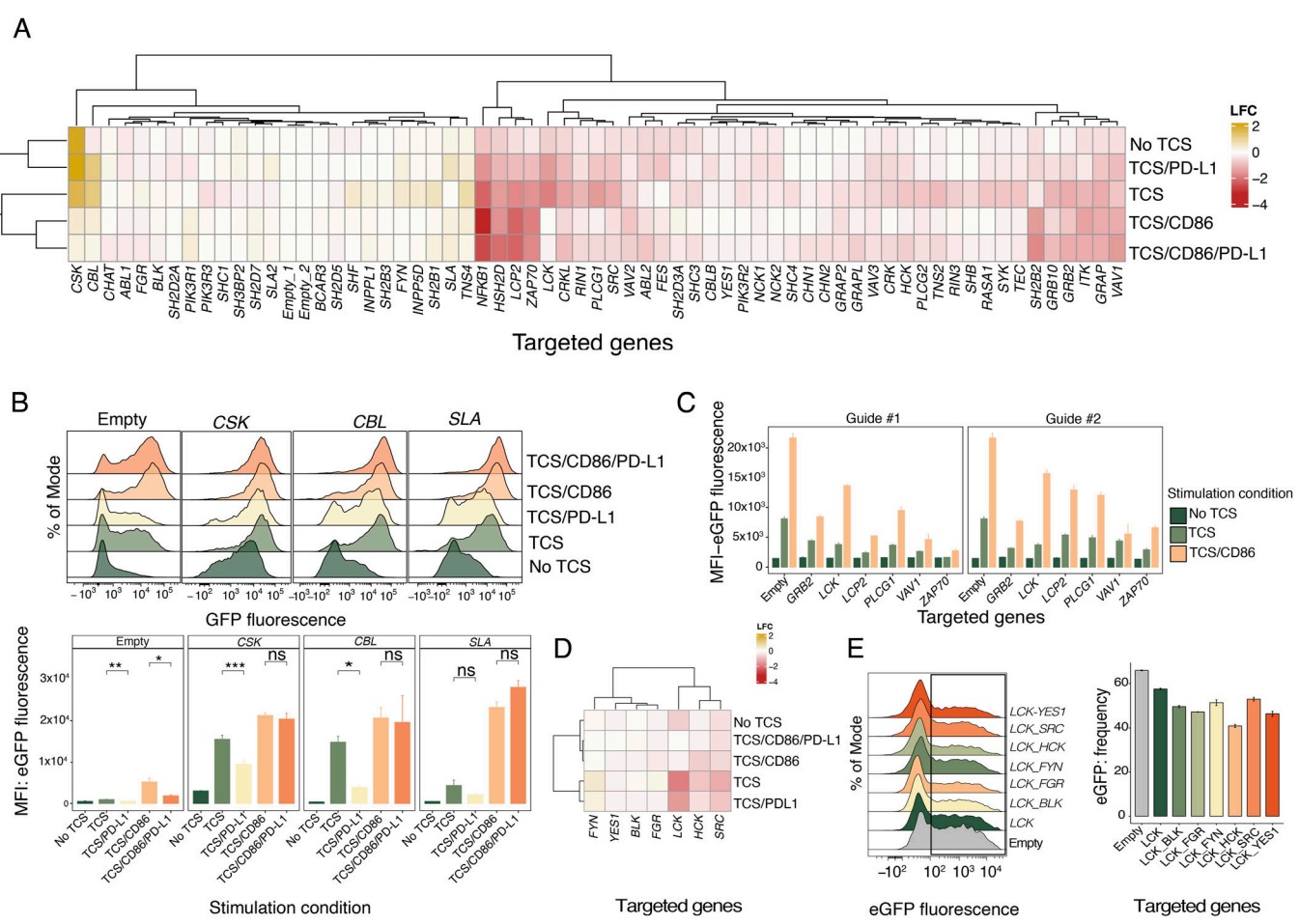

**Figure 2. Arrayed CRISPR KO screen of SH2 signaling proteins in the two-cell signaling assay.**
**(A)** Heatmap of the logarithm fold change of the eGFP median fluorescence intensity for each gRNA target and stimulation condition relative to the negative control (empty targeted). Data generated by calculating fold changes from mean MFIs from three replicates for each of the two separate targeting gRNAs. **(B)** eGFP fluorescence levels of Jurkat/PD-1 cells targeted with *CSK*, *CBL*, *SLA* or empty gRNA in five different T-cell stimulator (TCS) stimulation conditions (upper panel). eGFP fluorescence levels of Jurkat/PD-1 cells targeted with *CSK*, *CBL*, *SLA* or empty gRNA when unstimulated or stimulated in four different TCS stimulation conditions (upper panel). Pairwise comparisons between control and inhibition conditions performed using independent *t* test. **(C)** eGFP median fluorescence intensity of positive regulator (*GRB2*, *LCK*, *LCP2*, *PLCG1*, *VAV1*, *ZAP70*) gRNA-targeted Jurkat/PD-1 cells with two different gRNAs and under no stimulation or stimulation with TCS or TCS/CD86 using a low stimulation regimen. **(A, D)** Heatmap of logarithm fold change of the eGFP median fluorescence intensity for members of the SRC-family proteins (subset from (A)). **(E)** Fraction of cells expressing eGFP post stimulation with TCS/CD86 in Jurkat/PD-1 cells targeted with empty gRNA and wither *LCK* sgRNA or sgRNA targeting additional one other SRC family kinase sgRNA (*BLK*, *FGR*, *FYN*, *HCK*, *SRC*, *YES1*). Pairwise comparisons performed relative to *LCK*-only KO using one-way ANOVA. In all cases, the number of replicates is three and *P*-value asterisk represent: \*\*\**P* < 0.001, \*\**P* < 0.01, \**P* < 0.05; ns, not significant.

control), *ZAP70*, *LCP2*, *HSH2D*, *VAV1*, *GRB2*, *GRAP* and *GRB10*, *SH2B2*, *ITK*, *PLCG1*, *LCK*. In the heatmap distance clustering (Fig 2A), these genes clustered in two separate groups, prompting us to investigate whether this was because of the differences in gRNA efficiency, with one of the guides being more efficient, producing a smaller average effect for some of the targets. This was indeed the case. For all genes except *LCK* and *PLCG1*, a decrease in eGFP expression was observed under both signal 1 (TCS) and signal 1 + 2 (TCS/CD86) signaling contexts for at least one of the two gRNAs (Fig S3A and B). For the gRNAs with lower efficacy, we tested whether the signal 1 + 2 (TCS/CD86) conditions were masking the effects of the gRNAs by targeting representative genes (*ZAP70*, *LCP2*, *GRB2*, *VAV1*, *LCK*, *PLCG1*) using both gRNAs and reducing the degree of stimulation by decreasing the co-incubation time to 15 h and the Jurkat:TCS ratio from 3:1 to 10:1. Using this less-intense stimulating regimen, we observed

significant decreases in eGFP expression in both signal 1 (TCS) and signal 1 + 2 (TCS/CD86) stimulation conditions for all gRNAs (Figs 2C and S3C). Nevertheless, *LCK*-targeted cells still had higher residual eGFP expression when stimulated under signal 1 + 2 (TCS/CD86) conditions, even though the gRNAs targeting *LCK* dramatically reduced the levels of LCK at the protein level (Figs 2C and S3C and D).

LCK is an essential SRC–kinase responsible for initiating TCR signaling (Smith-Garvin et al, 2009), therefore it was unexpected that Jurkat cells targeted with *LCK*-specific gRNAs would have residual eGFP expression when stimulated under low-intensity signal 1 + 2 (TCS/CD86) conditions, whereas the same targets produced very little residual eGFP expression under signal 1 (TCS) conditions. This implies that co-stimulatory signaling by CD28 is especially LCK-independent. In our original screen of the six SRC family kinase members present (BLK, FGR, FYN, HCK, SRC, and YES-1), only SRC

targeting had a modest effect upon stimulation and only upon stimulation under signal 1 (TCS) conditions (Fig 2D). We therefore tested whether these kinases compensate for the absence of LCK in Jurkat T-cells. To test this, we targeted members of the SRC–kinase family in an *LCK*-targeted cell line and examined the responses of the doubly targeted cells to the lower intensity signal 1 + 2 (TCS/CD86) stimulation. Targeting each of these kinases in addition to *LCK* reduced eGFP expression, with the largest reduction seen with targeting *HCK* (Fig 2E). However, the effects were small, suggesting that all the kinases collectively compensate for the absence of LCK.

Overall, the arrayed screen offered a way to study the contributions of SH2 domain-containing proteins to signaling outcome under each of the five conditions tested. For many of the known positive and negative regulators, the assay produced the expected increases and decreases in eGFP expression. With respect to the identification of specific regulators of PD-1 inhibition, although we identified *CBL*, *CSK*, and *SLA* as targets whose KO influenced the potency of inhibitory signaling, we did not identify a new mediator of PD-1 signaling among the 64 SH2 domain-containing molecules.

## Genome-wide screening of positive and negative regulators of T-cell signaling

The arrayed screen indicated that there are no undiscovered mediators of PD-1 inhibitory signaling among the 64 SH2 domain-containing proteins expressed in Jurkat T cells. A genome-scale screen could, in principle, identify new potential mediators of PD-1 signaling outside of this group, in an unbiased manner. Genome-scale loss of function screens to identify the regulators of proximal T-cell signaling have been performed previously both in Jurkat and human primary T cells and such screens have provided valuable insights into known and novel regulators of T-cell signaling (Shang et al, 2018; Shifrut et al, 2018). These screens have, however, been performed in non-cell–based stimulation settings. Given that the two-cell system allows stimulation screens incorporating the engagement of inhibitory ligands, we undertook a genome-wide screen to identify regulators of T-cell signaling while PD-1 is engaged by PD-L1.

Our initial targeting of SHP1/2 in the arrayed CRISPR screen revealed that overstimulating cells can mask the effects of perturbing inhibitory signaling mediators, implying that stimulation via the TCR/CD3 complex alone (signal 1) would likely allow more sensitive identification of mutants that rescue signaling from PD-1 inhibition. Hence, we undertook genome-wide screens in the context of signaling by the TCR, and inhibitory signaling triggered by PD-1. We first conducted a genome-wide KO screen by transducing Jurkat/PD-1 cells with a lentiviral library encoding ~91,000 sgRNAs targeting ~18,000 genes and stimulating the mutant cells using TCS/PD-L1 13 d post-transduction. We collected mutant cells expressing high or low eGFP using flow cytometry, amplified the gRNAs present in these sorted populations, and then compared the gRNA counts with those of the control unsorted population. This identified gRNAs that were enriched in the two sorted populations (the sorting strategy is set out in Fig S4; screen data is available in Table S2). Cells expressing low levels of eGFP were enriched in gRNAs targeting positive regulators of TCR signaling as expected (Fig 3A, left panel). Similar to the Jurkat T-cell screen undertaken by Shang et al (2018) the hits from this

screen included gRNAs targeting genes encoding the components of the TCR–CD3 complex comprised of CD3-δ, CD3-ε, CD3-γ, and CD3-ζ (CD247) chains (the genome-wide library did not include gRNAs targeting *TCRA* and *TCRB*), the tyrosine kinases LCK, ZAP70, and ITK, and the LAT signaling complex, that is, LAT and LCP2 (Fig 3A, left panel). Enrichment analysis using the KEGG database revealed significant enrichment of immune-related signaling pathways including the TCR signaling pathway (Fig 3B). An interesting hit from this screen was *BAP1*, which is known to be essential for peripheral T-cell proliferation (Arenzana et al, 2018). There are reports that BAP1 regulates NFκB signaling in B lymphoblasts so, taken together, it is likely that BAP1 is involved in signaling leading to proliferation via the NFκB pathway in T cells (Takagi-Kimura et al, 2022).

We closely examined the genes that had false discovery rate (FDR) < 0.25 among the eGFP-high population of cells and categorised them according to cellular function (Table S3). We were interested especially in identifying signaling-related molecules that could mediate signaling by PD-1. Of the molecules involved in cellular signaling, the top hits included regulators of the NFκB-pathway (*NFKBIA*, *NFKB2*, *CYLD*, and *TNFAIP3*) and TNF receptor-associated factors including *TRAF2, 3* (Fig 3A, right panel). Pathway enrichment analysis also revealed a significant enrichment of the genes controlling the NFκB pathway (Fig 3B). We also identified previously characterised negative regulators of T-cell signaling such as *CBL* and *CSK*. Notably, the signaling regulators *PTPN11* (*SHP2*) and *SH3KBP1*, which has been shown to bind to CBL (Taipale et al, 2014), and *PDCD1*, the gene encoding PD-1 itself, were identified in the screen. The identification of *PDCD1* and *PTPN11* in this screen confirmed that this assay could identify mediators of PD-1-dependent inhibitory signaling. However, presumably because the consequent increases in activation after KO were smaller than those of general negative regulators, they were identified at a lower FDR. In general, the top negative regulator hits in this screen were mostly dominated by cellular factors whose targeting increased positive signaling to a very high level thereby reducing the power to detect mutants that more modestly enhance signaling, an effect we previously noted in the arrayed screen.

The genome-wide screen also revealed several hits in actin/cytoskeletal regulation pathways (Fig 3A, right panel). Among the top hits were *ARPC4* and *PFN1*, which encode one of seven subunits of the human Arp2/3 protein complex and an actin binding protein, respectively. *PFN1* is known to regulate NFκB pathway but also has been previously shown to be the only member of the profilin family to be expressed in primary human CD8[+] T cells where it acts as a negative regulator of cytotoxicity (Schoppmeyer et al, 2017). To further investigate the role of *ARPC4*, we designed gRNAs targeting this gene and tested the levels of activation post stimulation with the different TCS in both Jurkat/control and Jurkat/PD-1 lines. Knocking out *ARPC4* (Fig S5A) increased eGFP expression in both Jurkat/control and Jurkat/PD-1 in all stimulation contexts (Fig 3C). Together, these data suggest that *ARPC4* enhances T-cell signaling and is not a specific regulator of the PD-1 pathway.

We then tested if targeting *ARPC4* affected signaling responses in primary T cells. We targeted *ARPC4* in CD8[+] T cells and stimulated them using TCS/CD86 and measured the expression of two activation markers, CD69 and 4-1BB, after 24 h. We observed that targeting *ARPC4* had no effect on the expression of CD69 but

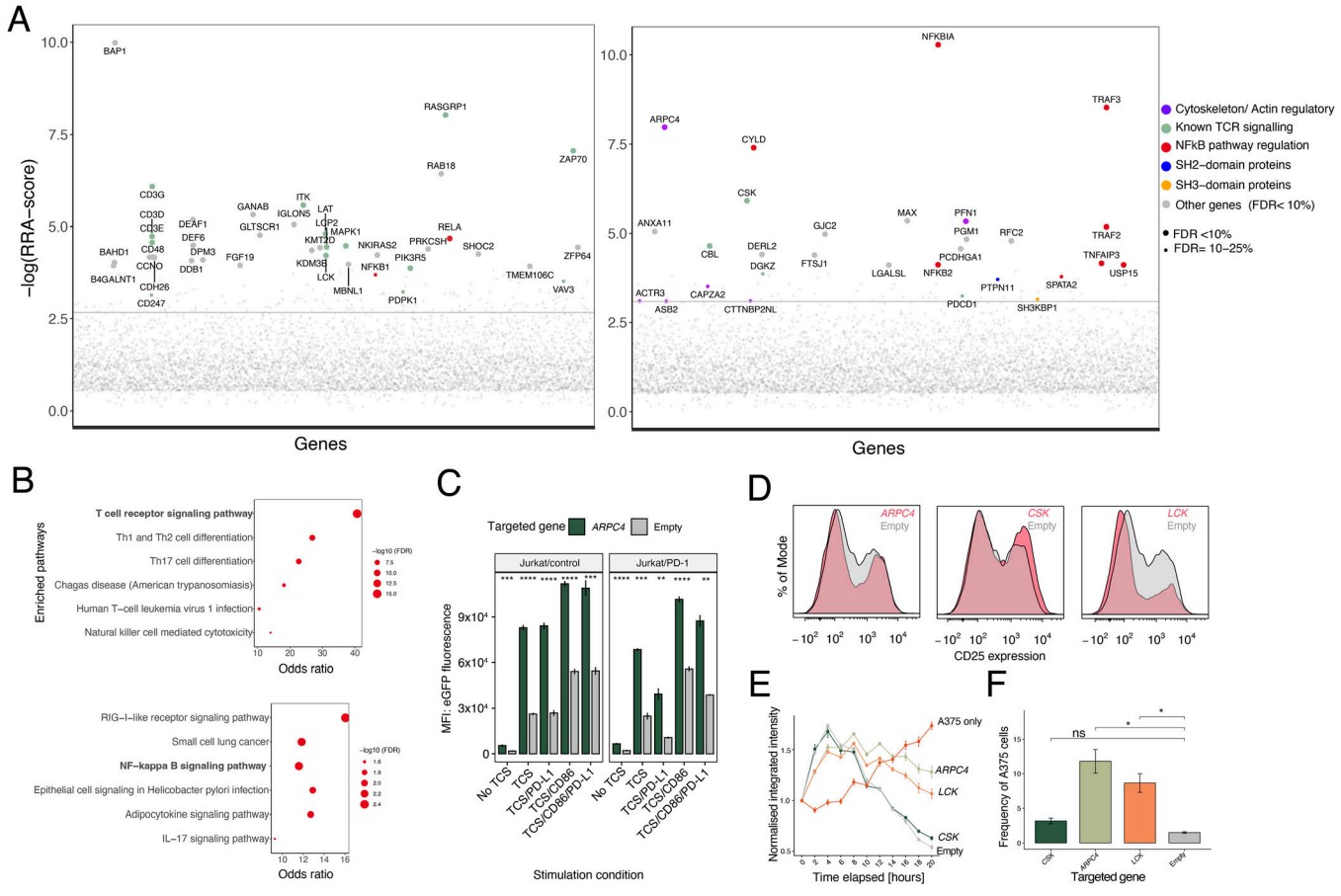

**Figure 3. A genome-wide pooled CRISPR screen of T cell activation and inhibition using the two-cell signaling assay.**
**(A)** The enrichment of gRNAs targeting each gene in cells with low eGFP (ordered alphabetically; left panel), and high eGFP (right panel) is quantified as the RRA-score. Each circle represents a specific gene. Only genes on the T-cell receptor signaling pathway and genes with false discovery rate < 10% are labelled and the green circles represent genes that are known to be related to the T cell function. Full data are available in Table S2. **(B)** Pathway enrichment of the hits from screen to identify positive regulators (low eGFP, upper panel) and negative regulators (high eGFP, lower panel). **(C)** Mean eGFP fluorescence intensity of parental Jurkat/control cells and Jurkat/PD1 cells targeted with *ARPC4* sgRNA co-cultured with four different T-cell stimulator conditions or cultured without stimulation. **(D)** Representative flow plots showing CD25 expression levels in *ARPC4*, *LCK* and *CSK*-targeted CD8⁺ cells compared with non-targeted control CD8⁺ cells, in the course of killing A375 target cells. **(E)** Changes in integrated red area intensity of A375 cells co-cultured with *ARPC4*, *LCK*, *CSK*, and empty-targeted CD8⁺ cells or cultured alone. **(F)** Percentage of A375 cells remaining after co-culture with targeted CD8⁺ cells for 20 h. Pairwise comparisons performed using one-way ANOVA, comparing with empty-targeted conditions. In (C, D, E, F), the number of replicates is three and P-value asterisk represent the following: ***P < 0.001, **P < 0.01, *P < 0.05; ns, not significant.

increased the expression of 4-1BB compared with parental CD8⁺ T cells reaffirming our previous findings in Jurkat cells (Fig S5B). We further tested the function of *ARPC4* in primary T cells using an in vitro cancer cell-killing assay. Primary CD8 T cells transduced to express the 1G4 TCR were used in a killing assay with an mOrange-expressing A375 melanoma cell line, which expresses the tumour antigen NY-ESO-1 (Chen et al, 2005). The level of killing was quantified by tracking the "integrated red area intensity" (see Fig S5C). ARPC4-targeted cells, and CSK-targeted (negative regulator control), *LCK*-targeted (positive regulator control), and empty guide-targeted 1G4-expressing T cells were generated and used in the killing assays (see Fig S5D for *ARPC4* KO efficiency). The *ARPC4* KO cells maintained and even slightly increased the level of expression of CD25 post interaction with A375 cell lines compared with the empty-targeted 1G4 TCR-expressing T cells (Fig 3D) but surprisingly had significantly reduced cytotoxic ability (similar to the levels of *LCK*-targeted cells; Fig 3E and F). Cells with disrupted actin pathways, whereas presenting as hyperactive

cells in terms of the expression of surface markers of activation, appear therefore to be profoundly impaired in terms of their effector functions.

# Discussion

We adapted a two-cell system used previously to compare the functions of immune checkpoints, to make it compatible with perturbation analysis, allowing the identification of T-cell signaling mediators. CRISPR-based activation screens typically rely on the activation of cells with soluble, plate-bound or bead-based anti-CD3 antibodies and on assessing activation via the expression of markers such as CD69. The cell-based activation system used here relied on stimulation by a surrogate APC, with the readout being the NFκB-regulated expression of eGFP, which integrates signaling from both activatory and inhibitory pathways in the reporter cell. We used the two-cell system in a genome-wide screen, the design of which

selected mutants with dysregulated TCR signaling, and mutant cells with high NFκB activity (indicative of strong signaling) despite the engagement of PD-1, thereby allowing the identification of mediators of inhibitory signaling by PD-1. The data showed that it is possible to perform a cell-based activation screen to identify positive regulators of T-cell signaling, creating opportunities for more nuanced signaling studies wherein the contribution of multiple receptors to T-cell signaling can be assessed in physiological settings. The design of the screen revealed how known negative regulators such as *CSK* and *CBL* affect signaling at different thresholds of activatory and inhibitory signaling. Targeting both genes resulted in increased signaling, thereby partially abrogating the effects of inhibition by PD-1 signaling.

The dominating effect of overstimulation on inhibitory signaling was a general theme of this study. In terms of the experimental approach, the genome-wide screening strategy used to identify regulators of PD-1 was impacted by the much larger effects of the global negative regulators versus those of specific mediators of inhibitory signaling (such as *PDCD1* itself and *PTPN11*), increasing their FDRs. The alternative approach of using an arrayed screen, which allows individual tests of signaling effects at different thresholds of activatory and inhibitory signaling, was better suited to identifying signaling molecules with weaker effects on signaling. Accordingly, targeting SHP2, a known regulator of PD-1 signaling, at least partially overcame PD-1-mediated inhibition in our arrayed screen. In our analysis of all the SH2 family proteins expressed by Jurkat T cells, the only other candidates found to have any role in mediating inhibitory signaling were the general negative regulators (*CBL*, *CSK*, *SLA*), suggesting that SHP2 is likely the major PD-1-specific, SH2 family mediator of inhibitory T-cell signaling. For now, the reason(s) why PD-1 retains residual inhibitory potential in T-cells lacking both SHP1 and SHP2 remains to be determined (Xu et al, 2020).

The new two-cell screening system reconfirmed the central roles of many previously characterised regulators of T-cell signaling, especially those proximal to the TCR. However, it also revealed several ways in which signaling in Jurkat T cells differ from that in primary cells. For example, both the genome-wide and arrayed screens identified CBL but not its paralog CBLB as negative regulator of T-cell activation, even though CBLB is a well-characterised negative regulator and is detected, for example, in screens of primary T cells (Shifrut et al, 2018). Interestingly, CBLB was not identified in a comparable screen performed using Jurkat cells by Shang et al (2018), and CBL was not identified in the primary T cell screen by Shifrut et al (2018), suggesting that the roles of CBLB and CBL have been swapped in Jurkats and primary T-cells. Similarly, FYN partially replaces the function of LCK in normal T cell development (Groves et al, 1996), but we did not identify a comparable role for FYN in Jurkat cells. These results serve to highlight the signaling differences between primary T cells and Jurkats.

We identified *ARPC4* as a key regulator of NFκB activation. The actin cytoskeleton has been studied in the context of T-cell activation and multiple roles have been ascribed to it, with one of the most important being its role in forming stable conjugates between T cells and APCs (Dustin & Cooper, 2000). The unexpected inhibitory role of actin was first revealed by Rivas et al (2004), who showed that treatment of T cells with cytochalasin D enhances cytokine production in stimulated Th1 and Th2 CD4⁺ T-cell clones and in CD4⁺ and CD8⁺ TCR transgenic T cells. It was proposed that these effects resulted from changes in the duration of TCR-induced Ca$^{2+}$/NFAT signaling (Rivas et al, 2004). We investigated this further and found that, although *ARPC4* KO increased signaling through proximal events up to the point of NFκB transcriptional activity, in primary cells, there is an overall decrease in cytotoxicity. This presumably arises because the cells are unable to deliver cytotoxic granules and perhaps generate higher-level, later-stage structures at the immunological synapse.

The contrasting effects of *ARPC4* KO serve to highlight the limitations of transformed cell line-based screening systems. With advancements in gene editing in primary cells, it should be possible to perform genome-wide screens of activatory and inhibitory signaling in settings in which the assays are not restricted to proximal activation read-outs. However, these assays are unlikely to be easy. We have observed that the expression of PD-1 in primary T cells is highly variable, and that the optimal window for observing inhibition is difficult to ascertain. PD-1 is not expressed in the resting state, and by the time it is expressed, the activation signal will likely be consolidated, reducing any inhibitory signaling effects. In addition, a degree of redundancy among co-expressed inhibitory receptors means that the contributions of individual receptors to inhibition will be hard to disentangle. The great advantage of cell lines, that is, the ease with which signaling receptors can be introduced exogenously, means that these systems will likely continue to be useful for unpicking signaling pathways in T cells, in ways illustrated by the present study.

# Materials and Methods

### Cell culture

Previously generated Jurkat NFκB::eGFP cell lines and BW cell lines expressing an anti-human CD3 single-chain fragment anchored by fusion to human CD14 (TCS cells) were used in this study (Leitner et al, 2010; Jutz et al, 2017). Jurkat cells and TCS cells were both cultured in an RPMI 1640 medium (Gibco) supplemented with 10% (vol/vol) FBS (lot #08F6480K; Gibco), 1% (vol/vol) penicillin/streptomycin/neomycin (final concentrations 50 U/ml penicillin, 50 µg/ml streptomycin, 100 µg/ml neomycin; Sigma-Aldrich) and 1% Hepes–NaOH (final concentration 10 mM; Sigma-Aldrich). HEK-293T cells for lentiviral transfection were cultured in DMEM (Gibco), supplemented with 10% (vol/vol) FBS (lot #08F6480K; Gibco), 1% (vol/vol) penicillin/streptomycin/neomycin (final concentrations 50 U/ml penicillin, 50 µg/ml streptomycin, 100 µg/ml neomycin; Sigma-Aldrich), 1% (vol/vol) sodium pyruvate (final concentration 1 mM; Sigma-Aldrich), 1% (vol/vol) L-glutamine (final concentration 2 mM; Sigma-Aldrich) and 1% (vol/vol) Hepes–NaOH (final concentration 10 mM; Sigma-Aldrich).

### Two-cell assays

The two-cell assay was performed as described previously (Jutz et al, 2017). Briefly, Jurkat cells were cultured in a 3:1 ratio with TCS cells expressing high levels of CD86 and PD-L1. CD86 and PD-L1 binding was blocked using 5 µg/ml Ultra-LEAF anti-human CD86 and PD-L1 antibodies (Clone #IT2.2 and #29E.2A3; BioLegend,

respectively) to create different stimulation conditions. After 24 h, cells were harvested and TCS cells stained with PE-conjugated anti-mouse CD45 antibody (Clone #I3/2.3; BioLegend). Expression of eGFP reporter gene in Jurkat cells was measured by flow cytometry.

## sgRNA design and cloning

sgRNA sequences used were the top two candidates from CRISPick (Broad Institute) for each target gene. sgRNA sequences, as oligos, (Sigma-Aldrich) were annealed and cloned into the BbsI site of the CRISPR gRNA expression vector pKLV2-U6gRNA5(BbsI)-PGKpuro2ABFP-W (#67974; Addgene) as described before (Sharma & Wright, 2020). The full list of gRNAs for all the targets used in this study are listed in Table S4.

## Lentiviral preparation

Lentiviruses were prepared by transfecting HEK-293T cells with the gRNA expression vector. 500 ng of gRNA expression plasmid DNA was incubated with 250 ng psPAX2 (#12260; Addgene) and 150 ng psMD2.G (#12259; Addgene) and 3 μl of PLUS reagent (Invitrogen) in 250 μl OptiMEM (Gibco). After 5 min, 3 μl Lipofectamine LTX (Invitrogen) was added to the mixture. The mixture was added to confluent HEK-293T cells in a 12-well plate. After 3 h of incubation, the OptiMEM mixture was removed and replaced with DMEM (Gibco), supplemented as described above. Lentiviral supernatant was harvested and filtered at 0.45 μm after 48 h. Viral supernatants were stored at –80°. This protocol has also been described in detail before (Sharma & Wright, 2020).

## Arrayed screening

Jurkat cells ($1 \times 10^5$ cells/well in 96-well plate) were transduced with the arrayed library lentiviruses followed by centrifugation at 800$g$ for 90 min. After 2 d, transduced cells were selected by addition of puromycin (final concentration 2 μg/ml; Gibco). After 7–10 d of growth, TCS cells ($5 \times 10^4$/well in a 96-well plate) were added to Jurkat cells, with different conditions created as described above. After 24 h, cells were harvested and eGFP expression was measured by flow cytometry. Jurkat cells were gated in flow cytometry using their BFP expression.

## Lentiviral transduction for generation of Cas9 lines

Cas9-expressing Jurkat/control and Jurkat/PD-1 lines were generated by lentiviral transduction of pKLV2-EF1a-Cas9Bsd-W construct. Cas9 activity in the cells was tested using a "BFP-GFP" system described before (Sharma et al, 2018). Briefly, cells were transduced with lentivirus encoding GFP, BFP, and a gRNA targeting GFP (pKLV2-U6gRNA5(gGFP)-PGKBFP2AGFP-W) or the same construct with an "empty" gRNA (pKLV2-U6gRNA5(Empty)-PGKBFP2AGFP-W) as a negative control. The activity of Cas9 in the cell lines was assessed by examining the ratio of BFP only to GFP-BFP–double-positive cells transduced by the two lentiviruses.

## Genome-wide screens

Human Improved Genome-wide Knockout CRISPR Library v1 was a gift from Kosuke Yusa (#67989; Addgene) (Tzelepis et al, 2016). Genome-scale screens were performed as described in a detailed protocol before (Sharma & Wright, 2020). Detailed protocol with all PCR steps, reaction times, and volumes are described in this protocol. Briefly, a genome-scale "knockout" library of Jurkat/PD-1 cells was produced by transducing 80 million cells such that ~30% of the total cell population was transduced. The transduced (BFP-positive) cells were harvested 3 d after transduction using a cell sorter (MA900; Sony sorter). Libraries with at least $1.5 \times 10^8$ cells were selected and maintained in media containing 2 μg/ml puromycin to remove the non-transduced cells and at every passage. 13 d post-library generation, 150 million cells were activated with TCS/PD-L1 for 24 h. Two-way sorting was performed to collect the top 15% and bottom 15% of eGFP expressing cells. Genomic DNA was extracted using a commercial kit (Blood and Tissue mini kit) and gRNA was amplified using the isolated DNA using L1/U1 primers and Q5 Hot Start High-Fidelity 2 × Master Mix. The PCR-amplified guides were diluted to 40 pg/μl in EB and tagged with Illumina index primers using second round of PCR with using 200 pg template, PE 1.0 as forward primer, appropriate index tags as reverse primers, and KAPA HiFi HotStart ReadyMix polymerase. The PCR products were cleaned using SPRI beads (Agencourt AMPure XP beads), quantified using qbit and pooled to 4 nM concentration and sequenced using NextSeq platform. Single-end sequencing (19 bps) was performed with the custom sequencing primer, 5'-TCTTCCGATCTCTTGTGGAAAGGACGAAACACCG-3'.

## Data analysis

The read count for each gRNA and gene level enrichment analysis was carried out using the MAGeCK statistical package (version, v0.5.5) by comparing the read counts from the sorted population with those from the control population. Additional parameters in MAGeCK were used to only include genes that had more than 3/5 guides enriched in each test ("min-number-goodsgrna 3"). Pathway analysis was also carried out using the EnrichR package with KEGG annotated pathways. All further analyses were carried out using R.

## Generation of 1G4-T cells

PBMCs from healthy blood donors were activated with anti-CD3/anti CD28 beads for 2 d. The cells were then transduced with a lentiviral construct encoding the 1G4 construct on a retronectin plate. The 1G4 construct consisted of a short LNGFR fragment. Transduced 1G4-expressing CD8 cells were isolated using FACS sorting of LNGFR+/CD8+ population. Cells were proliferated for 2 wk after which they were electroporated using the RNP complex to generate KO cells. For the generation of RNPs, commercially available synthetic gRNAs (from Synthego) were preincubated at room temperature with Cas9 (IDT; Alt-R S.p. Cas9 Nuclease) (75 pmol gRNA: 31 pmol Cas9) for 30 min. One million cells per reaction were resuspended in 20 μl P3 buffer (Lonza) as per the manufacturer's instructions. The RNP complex was added to the cells and electroporated on a 16-well strip using Lonza electroporator with EO115 setting. 80 μl of cells were added immediately to the cells and

allowed to recover for 10 min. Cells were then plated on 2 wells of 96 wells with 150 μl media in each well. The mutant cells were used 7–10 d post electroporation.

**In vitro A375 cell-killing assay with 1G4-T cells**

A375 cells were engineered to constitutively express mOrange via lentiviral transduction. For setting the killing assay, 30,000 A375 cells were plated per well on a 96-well plate for 24 h before adding 90,000 (1:3 ratio) of T cells. The T cells used for this purpose were cultured in media without IL-2 for 24 h before use on killing assays. The plate was imaged using IncuCyte for 20 h after which all the cells from the wells were extracted. Leftover A375 cells were extracted using trypsin. The T cells were stained with anti-CD25 antibody to assess the level of activation. Killing was determined by assessing the percentage of leftover A375 cells post 20 h of co-incubation with 1G4 T cells. Killing levels were also quantified by measuring the area and the intensity of "red" fluorescence.

**Assessment of KO efficiency**

1G4-TCR and Jurkat KO of ARPC4 were validated using an antibody against ARPC4 Proteintech rabbit anti-human ARPC4, polyclonal antibody (Cat no. 10930-1-AP). *LCK* KO Jurkat cells were validated using anti-LCK antibody (Clone 3A5).

# Supplementary Information

# Acknowledgements

This research was funded by Wellcome Trust [Grant 207547/Z/17/Z to SJ Davis and Grant 215883/Z/19/Z to S Sharma]. For the purpose of Open Access, the author has applied a CC BY public copyright licence to any Author Accepted Manuscript (AAM) version arising from this submission. We would like to thank the WIMM flow cytometry core facility and WIMM sequencing facility for assistance with processing CRISPR-screening samples.

## Author Contributions

S Sharma: conceptualization, formal analysis, supervision, funding acquisition, validation, investigation, visualization, methodology, project administration, and writing—original draft, review, and editing.
T Whitehead: formal analysis, validation, investigation, visualization, methodology, and writing—review and editing.
M Kotowski: investigation, methodology, and writing—review and editing.
EZQ Ng: validation, investigation, and methodology.
J Clarke: methodology.
J Leitner: methodology.
Y-L Chen: investigation and methodology.
AM Santos: investigation and methodology.
P Steinberger: investigation, methodology, and writing—review and editing.
SJ Davis: conceptualization, supervision, funding acquisition, and writing—review and editing.

## Conflict of Interest Statement

The authors declare that they have no conflict of interest.

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
