## [Reviewer comments · Life Science Alliance]

Life Science Alliance

A high-throughput two-cell assay for interrogating inhibitory signaling pathways in T cells.

Sumana Sharma, Toby Whitehead, Mateusz Kotowski, Emily Ng, Joseph Clarke, Judith Leitner, Yi-Ling Chen, Ana Mafalda Santos, Peter Steinberger, and Simon Davis

DOI: <https://doi.org/10.26508/lsa.202302359>

Corresponding author(s): Sumana Sharma, University of Oxford and Simon Davis, University of Oxford

Review Timeline:

Submission Date:	2023-09-08
Editorial Decision:	2023-10-25
Revision Received:	2023-11-27
Editorial Decision:	2023-11-28
Revision Received:	2023-11-30
Accepted:	2023-11-30

Transaction Report:

October 25, 2023

Re: Life Science Alliance manuscript #LSA-2023-02359-T

Sumana Sharma
MRC Human Immunology Unit, University of Oxford
Headley Way
Oxford OX3 9DU
United Kingdom

Dear Dr. Sharma,

Thank you for submitting your manuscript entitled "A high-throughput two-cell assay for interrogating inhibitory signaling pathways in T cells." to Life Science Alliance. The manuscript was assessed by expert reviewers, whose comments are appended to this letter. We invite you to submit a revised manuscript addressing the Reviewer comments.

Thank you for this interesting contribution to Life Science Alliance. We are looking forward to receiving your revised manuscript.

Sincerely,

B. MANUSCRIPT ORGANIZATION AND FORMATTING:

Reviewer #1 (Comments to the Authors (Required)):

The study is based on a 2- cell ("APC" and T cell) system used to screen positive and negative regulators of T cell signalling using a Jurkat reporter cell line as the T cell. The system is then used to test an arrayed CRISPR/Cas screen a of SH2 domain proteins and their role in T cell signalling in the presence of both CD28 and PD-1 signalling. This identified genes such as SLA, CSK etc. In a follow up series of experiments the authors tested the system in a genome wide CRISPR screen.

This is a well conducted and well written report, which is confirmatory rather than revolutionary in its conclusions. Nonetheless, I have no real criticisms of the work and feel it forms a useful addition to the literature and provides an important resource for others working in this area. It can be published as is. My only minor comment is that the nature of the APC could be a bit more explicit in the text.

Reviewer #2 (Comments to the Authors (Required)):

In this manuscript, Sharma et al. built two CRISPR/Cas9-based functional genetic screening systems using a two-cell assay comprising Jurkat reporter cells and T cell stimulator cells. The screens were performed in both arrayed and pooled formats, leading to the identification of known mediators of both activator and inhibitory signals in T cells, demonstrating the utility of the screening systems. Furthermore, the authors discussed a discrepancy in the functions of ARPC4 between T cell lines and primary T cells separately co-cultured with TSC cells and target tumor cells.

Overall, the data is well-documented and the manuscript is well-written. I have only one minor comment to make:

It is worth noting that the kinetics of T cell activation markers, such as CD25, CD69, CD137, IL-2, IFN γ , and TNF α , can vary. Therefore, comparing several markers to determine the activation status and extent could enhance the robustness of the data. So additional T cell activation markers should be assessed for figures 3C and 3D to provide a more comprehensive analysis.

Reviewer #3 (Comments to the Authors (Required)):

The manuscript by Sharma et al describes the development of a novel high-throughput two-cell assay for interrogating inhibitory signaling pathways in T cells. As the authors state, immunotherapies targeting T cells are one of the most exciting and promising areas of recent drug advancement and reflect highly dynamic areas of academic research and drug development activity. Despite undoubted successes there is still much we do not know about generating and optimising agents targeting these potent adaptive immune cells. Generating high-throughput and effective cell-based assays to determine the impact and importance of costimulatory or inhibitory signalling to the efficacy of T cell directed approaches is an area of clear need. The authors here present very clear and robust data supporting their claims to have generated such an assay. The data presented is appropriately controlled, contains sufficient replicates to demonstrate robustness and is supported by statistical significance. Overall, this manuscript is well put together and will provide an excellent resource for those seeking to undertake such assays in the future.

There are however, some minor issues with regards to wording and clarity of both the text and the figures presented. All of these minor manuscript issues, to this reviewer's mind, can readily be corrected on revision. Experimentally, the work is well executed and the data robust.

Minor comments:

Author affiliation details provided are not numbered to correspond to numbers with author list.

The word 'activatory' is used in the abstract and throughout the manuscript, this word does not exist to my knowledge and in any dictionaries, I can locate readily and so should be replaced with 'activating'.

There are places where the language used is not scientific or precise. The manuscript would benefit from a careful proofread for language and grammar use. E.g. In the introduction the authors make the statement. "Heavy TCR/CD3 complex phosphorylation allows.....". Heavy is not the best wording to use. There are other examples of non-scientific wording that should be changed and a few typos that could be corrected.

At the end of the introductory section the authors state that, "Here, we first adapt the cellular assay developed by...." Can the authors briefly say how they adapted the assay here? Would help the reader follow what is to come.

In the results section, the authors introduce their T-cell stimulator cells (TCS) could the authors briefly describe these at first introduction so the non-specialist reader does not need to go away and find out what these are.

The first set of results text describing the data presented in figure 1 is very dense and difficult to follow. Could the authors break this down a little and refer to figure panels more extensively to take the reader through this data. The data is solid but it is difficult to follow as described. The use of terminology such as Signal 1 and Signal 1+2 etc when the figure legend refers to TCS/CD86 etc is very confusing. Could the authors common up their nomenclature to make clearer. It might be better to add in (TCR/CD86) etc alongside Signal 1+2 in text.

The figures themselves have numerous small panels that are difficult to read and interpret easily. This is a pity as the authors present great data. Can the authors maximise the size of figures on the page (e.g. don't leave big white areas around figures etc). The panels need more and clearer labelling and better linking to the text. Not sure if this is journal style, but listing the legend takes up more space than necessary. These comments apply to all figures.

When discussing figure 2, the authors state that, "We compiled a list of 64 SH2..... based on transcriptome analysis." Could the authors provide references or a statement as to the source of this data? Again, the writing of this section is dense and complex and requires the reader to keep moving back and forth between text and figure. It would be helpful if the authors could move the figure closer to the text that describes it. This applies to all figures.

When taking the reader through figure 3 the authors have a sentence saying, "We collected mutant cells....." In this sentence the authors use very repetitious language ('sorted' features in various forms 3 times in this sentence. Need to reword.

When discussing the final figure, the authors make a point about the sensitivity of their assay. Can the authors comment on what the affinity of the scFv used as TCR engager was and whether this could impact responses observed and how physiological this might be?

The sentence, "We further tested te function of ARPC4....." Is not a complete sentence.

Again, in the discussion the authors need to make sure they are using appropriate language. E.g. "For now, the reason(s)..... remains a mystery...." Perhaps '.... remains to be determined.' might be better?

Overall this is an excellent piece of work describing an assay that could serve the field well, but it would benefit from being clearer and easier to read.

Response to reviewers

Reviewer #1

The study is based on a 2- cell ("APC" and T cell) system used to screen positive and negative regulators of T cell signalling using a Jurkat reporter cell line as the T cell. The system is then used to test an arrayed CRISPR/Cas screen a of SH2 domain proteins and their role in T cell signalling in the presence of both CD28 and PD-1 signalling. This identified genes such as SLA, CSK etc. In a follow up series of experiments the authors tested the system in a genome wide CRISPR screen.

This is a well conducted and well written report, which is confirmatory rather than revolutionary in its conclusions. Nonetheless, I have no real criticisms of the work and feel it forms a useful addition to the literature and provides an important resource for others working in this area. It can be published as is. My only minor comment is that the nature of the APC could be a bit more explicit in the text.

Thank you for the positive comments on our work. The APCs we used are previously published edited BW cell lines (Leitner *et al.* 2010). We set this out in the Methods section but we agree that it is better placed in the main text. We have edited the introduction to make the meaning of TCS clearer.

"An assay of this type has previously been established by Jutz *et al.*, who used it to characterize the functions of both activatory and inhibitory receptors (Jutz *et al.*, 2017) using fluorescence-based transcriptional reporters expressed in the human Jurkat T-cell line. In these assays, a T-cell stimulator (TCS) cell, which is a murine thymoma-derived cell line (BW5417) engineered to express a membrane-bound single-chain Fv (scFv) variant of the anti-CD3 ϵ antibody OKT3, is used as the APC (Leitner *et al.*, 2010). TCS cells are versatile tools as they can be readily engineered to express a given surface protein to study T-cell coinhibitory or costimulatory processes, hence this system provides an ideal platform to characterise signalling processes in T cells."

Reviewer #2

In this manuscript, Sharma *et al.* built two CRISPR/Cas9-based functional genetic screening systems using a two-cell assay comprising Jurkat reporter cells and T cell stimulator cells. The screens were performed in both arrayed and pooled formats, leading to the identification of known mediators of both activator and inhibitory signals in T cells, demonstrating the utility of the screening systems. Furthermore, the authors discussed a discrepancy in the functions of ARPC4 between T cell lines and primary T cells separately co-cultured with TSC cells and target tumor cells.

Overall, the data is well-documented and the manuscript is well-written. I have only one minor comment to make:

It is worth noting that the kinetics of T cell activation markers, such as CD25, CD69, CD137, IL-2, IFN γ , and TNF α , can vary. Therefore, comparing several markers to determine the activation status and extent could enhance the robustness of the data. So additional T cell activation markers should be assessed for figures 3C and 3D to provide a more comprehensive analysis.

Thank you for the positive reaction to our work.

We agree that the kinetics of T cell markers can vary and as we have shown the activation markers do not necessarily correspond to the level of effector function (target cell killing). We have seen that the KO

of *ARPC4* in primary cells leads to nominal differences in surface marker expression such as CD25 but are killing impaired, which we have shown in Figure 3D.

Originally in one of our early experiments we observed that the level of IL2 produced by CD3⁺ cells in response to PMA was elevated in *ARPC4* targeted cells (Figure 1A, below). This was consistent with a report from Rivas et al. 2016. However, we wondered if the defect in actin would prevent cytokine release and what we observed was accumulation of IL2 inside of the cells rather than increased production. Hence, we went on to look for other markers and functional assays to define the role of *ARPC4* in primary cells. Many of these markers are not robustly expressed in Jurkat cells and so we have now performed additional experiments in primary cells to show the level of 41BB and CD69 on primary T-cells. We again saw that there was little difference in the expression of CD69 while comparing the *ARPC4* targeted cells with the parental cells (Figure 1B, below). The expression of 41BB on the other hand was elevated in *ARPC4* targeted cells compared to control cells (Figure 1B). This again reiterates our point that *ARPC4* cells are not signalling deficient (if anything, the activation markers suggest they are more proficient) but are highly killing deficient.

We will now include Figure 1B in the supplementary section and add the following sentence in the main manuscript. We have decided to leave the PMA data out (Figure 1A) as this method does not engage the TCR and would not be meaningful for this section.

Figure 1: Assessment of T-cell activation markers on *ARPC4* targeted primary T-cells.

- A. Expression of IL2 on CD3+ parental cells or CD3+ *ARPC4* targeted cells post treatment with PMA for 4 hours.
- B. Expression of 41BB and CD69 on CD8+ parental cells or CD8+ *ARPC4* targeted cells post activation with TCS/CD86 for 24 hours.

The following is now added into the main manuscript:

“We then tested if targeting *ARPC4* affected signaling responses in primary T-cells. We targeted *ARPC4* in CD8⁺ T cells and stimulated them using TCS/CD86 and measured the expression of two activation markers, CD69 and 4-1BB, after 24 hours. We observed that targeting *ARPC4* had no effect on the expression of CD69 but increased the expression of 4-1BB compared to parental CD8⁺ T cells reaffirming our previous findings in Jurkat cells (Supplementary Fig. 5B).”

Reviewer #3

The manuscript by Sharma et al describes the development of a novel high-throughput two-cell assay for interrogating inhibitory signaling pathways in T cells. As the authors state, immunotherapies targeting T cells are one of the most exciting and promising areas of recent drug advancement and reflect highly dynamic areas of academic research and drug development activity. Despite undoubted successes there is still much we do not know about generating and optimising agents targeting these potent adaptive immune cells. Generating high-throughput and effective cell-based assays to determine the impact and importance of costimulatory or inhibitory signalling to the efficacy of T cell directed approaches is an area of clear need. The authors here present very clear and robust data supporting their claims to have generated such an assay. The data presented is appropriately controlled, contains sufficient replicates to demonstrate robustness and is supported by statistical significance. Overall, this manuscript is well put together and will provide an excellent resource for those seeking to undertake such assays in the future.

There are however, some minor issues with regards to wording and clarity of both the text and the figures presented. All of these minor manuscript issues, to this reviewer's mind, can readily be corrected on revision. Experimentally, the work is well executed and the data robust.

Minor comments:

Author affiliation details provided are not numbered to correspond to numbers with author list.

Thank you for your generous response to our study. And thank you for pointing out the issue with the affiliations. We have now amended this.

The word 'activatory' is used in the abstract and throughout the manuscript, this word does not exist to my knowledge and in any dictionaries, I can locate readily and so should be replaced with 'activating'.

Thank you for this comment, but we respectfully disagree. We find that 'activatory' is often used, particularly pertaining to T-cell receptor/signal types (see examples: Speiser et al, Journal of Immunology 2001, Spodzieja et al, PloS One 2017, Marlin et al. PloS One 2012). Searching Pubmed with "activatory" produces >800 hits. We also found this word listed in the Collins English dictionary with specific reference to cellular signalling: (see <https://www.collinsdictionary.com/dictionary/english/activatory>). Whilst we note that its usage has decreased in recent years, for consistency with previous work we would prefer to maintain its usage throughout the manuscript.

There are places where the language used is not scientific or precise. The manuscript would benefit from a careful proofread for language and grammar use. E.g. In the introduction the authors make the statement. "Heavy TCR/CD3 complex phosphorylation allows.....". Heavy is not the best wording to use. There are other examples of non-scientific wording that should be changed and a few typos that could be corrected.

Thank you for this comment. We have checked the manuscript for errors and changed instances of imprecise language. We have deleted the word "heavy" at this particular instance.

At the end of the introductory section the authors state that, "Here, we first adapt the cellular assay developed by...." Can the authors briefly say how they adapted the assay here? Would help the reader follow what is to come.

The adaptation relates to generating a Cas9 expressing version of the reporter line to make it compatible for large-scale genomic perturbation. We have now changed the sentence to the following:

"Here, we first adapt the cellular assay developed by Jutz et al., by generating a Cas9 expressing version of the Jurkat cell reporter line to make it amenable to systematic arrayed and large-scale CRISPR-based "knock-out (KO)" screens."

In the results section, the authors introduce their T-cell stimulator cells (TCS) could the authors briefly describe these at first introduction so the non-specialist reader does not need to go away and find out what these are.

We have now added a fuller explanation of the TCS line in the introduction:

"An assay of this type has previously been established by Jutz et al., who used it to characterize the functions of both activatory and inhibitory receptors (Jutz *et al*, 2017) using fluorescence-based transcriptional reporters expressed in the human Jurkat T-cell line. In these assays, a T-cell stimulator (TCS) cell, which is a murine thymoma-derived cell line (BW5417) engineered to express a membrane-bound single-chain Fv (scFv) variant of the anti-CD3 ϵ antibody OKT3, is used as the APC (Leitner *et al*, 2010). TCS cells are versatile tools as they can be readily engineered to express a given surface protein to study T-cell coinhibitory or costimulatory processes, hence this system provides an ideal platform to characterise signalling processes in T cells."

The first set of results text describing the data presented in figure 1 is very dense and difficult to follow. Could the authors break this down a little and refer to figure panels more extensively to take the reader through this data. The data is solid but it is difficult to follow as described. The use of terminology such as Signal 1 and Signal 1+2 etc when the figure legend refers to TCS/CD86 etc is very confusing. Could the authors common up their nomenclature to make clearer. It might be better to add in (TCR/CD86) etc alongside Signal 1+2 in text.

Thank you for this comment. Given the many different contexts we examined, we agree that there might be confusion in the naming. Accordingly, we have now indicated which TCS was used next to which signals were generated in the figure.

The figures themselves have numerous small panels that are difficult to read and interpret easily. This is a pity as the authors present great data. Can the authors maximise the size of figures on the page (e.g. don't leave big white areas around figures etc). The panels need more and clearer labelling and better linking to the text. Not sure if this is journal style, but listing the legend takes up more space than necessary. These comments apply to all figures.

This is a very fair suggestion. We have made the figures larger and removed the white spaces. We will also ask the journal to format the figures such that they are large when the manuscript is type-set.

When discussing figure 2, the authors state that, "We compiled a list of 64 SH2..... based on transcriptome analysis." Could the authors provide references or a statement as to the source of this data?

The transcriptomic data is from a cell model passport (<https://cellmodelpassports.sanger.ac.uk/>), which provides RNA-seq information for Jurkat cell lines. The expression values for the SH2 family members used in the paper are now provided in Supplementary Table 1. We have added the citation in the main text.

Again, the writing of this section is dense and complex and requires the reader to keep moving back and forth between text and figure. It would be helpful if the authors could move the figure closer to the text that describes it. This applies to all figures.

We hope that in the final version, the figure links will be 'clickable' so that the corresponding figures will come into view as soon as it is mentioned in text. But thank you for pointing this out as we will enquire about this.

When taking the reader through figure 3 the authors have a sentence saying, "We collected mutant cells....." In this sentence the authors use very repetitious language ('sorted' features in various forms 3 times in this sentence. Need to reword.

This has been reworded in the following way which we hope the reader will find more elegant and clearer.

"We collected mutant cells expressing high or low eGFP using flow cytometry, amplified the gRNAs present in these sorted populations, and then compared the gRNA counts with those of the control unsorted population. This identified gRNAs that were enriched in the two sorted populations (the sorting strategy is set out in Supplementary Fig. 4; screen data is available in Supplementary Table 2)."

When discussing the final figure, the authors make a point about the sensitivity of their assay. Can the authors comment on what the affinity of the scFv used as TCR engager was and whether this could impact responses observed and how physiological this might be?

The TCS used in the assay engages the TCR through scFv of OKT3. OKT3 binds to CD3 with a K_d in the μM range (2.63 μM ; Kjer-Nielsen et al. PNAS, 2004), which is comparable to that of physiological ligands (pMHC). We therefore expect the T-cell responses we measured to be highly physiologically relevant.

The sentence, "We further tested te function of ARPC4....." Is not a complete sentence.

This has been corrected in the manuscript.

Again, in the discussion the authors need to make sure they are using appropriate language. E.g. "For now, the reason(s)..... remains a mystery...." Perhaps '.... remains to be determined.' might be better? This has been rewritten as suggested, and the rest of the discussion has been proofread and care taken to use appropriate language.

Overall this is an excellent piece of work describing an assay that could serve the field well, but it would benefit from being clearer and easier to read.

Thank you once again for this response.

November 28, 2023

RE: Life Science Alliance Manuscript #LSA-2023-02359-TR

Dr. Sumana Sharma
University of Oxford
Headley Way
Oxford, Oxford OX3 9DU
United Kingdom

Dear Dr. Sharma,

Thank you for submitting your revised manuscript entitled "A high-throughput two-cell assay for interrogating inhibitory signaling pathways in T cells.". We would be happy to publish your paper in Life Science Alliance pending final revisions necessary to meet our formatting guidelines.

- please add ORCID ID for the secondary corresponding author--they should have received instructions on how to do so
- it is recommended to remove figures from the manuscript text. They should be uploaded separately, and their legends should be provided after the references section
- please add an Author Contributions section to your main manuscript text
- please add your main, supplementary figure, and table legends to the main manuscript text after the references section
- please add callouts for Figure S3B, D to your main manuscript text

Figure Checks:

- please add sizes next to the blots in Figure S5

A. FINAL FILES:

B. MANUSCRIPT ORGANIZATION AND FORMATTING:

Sincerely,

November 30, 2023

RE: Life Science Alliance Manuscript #LSA-2023-02359-TRR

Dr. Sumana Sharma
University of Oxford
Headley Way
Oxford, Oxford OX3 9DU
United Kingdom

Dear Dr. Sharma,

Thank you for submitting your Methods entitled "A high-throughput two-cell assay for interrogating inhibitory signaling pathways in T cells.". It is a pleasure to let you know that your manuscript is now accepted for publication in Life Science Alliance. Congratulations on this interesting work.

DISTRIBUTION OF MATERIALS:

Again, congratulations on a very nice paper. I hope you found the review process to be constructive and are pleased with how the manuscript was handled editorially. We look forward to future exciting submissions from your lab.

Sincerely,
